# Customizing Text-to-Image Generation with Inverted Interaction

## ABSTRACT

Subject-driven image generation, aimed at customizing user-specified subjects, has experienced rapid progress. However, most of them focus on transferring the customized appearance of subjects. In this work, we consider a novel concept customization task, that is, capturing the interaction between subjects in exemplar images and transferring the learned concept of interaction to achieve customized text-to-image generation. Intrinsically, the interaction between subjects is diverse and is difficult to describe in only a few words. In addition, typical exemplar images are about the interaction between humans, which further intensifies the challenge of interaction-driven image generation with various categories of subjects. To address this task, we adopt a divide-and-conquer strategy and propose a two-stage interaction inversion framework. The framework begins by learning a pseudo-word for a single pose of each subject in the interaction. This is then employed to promote the learning of the concept for the interaction. In addition, language prior and cross-attention loss are incorporated into the optimization process to encourage the modeling of interaction. Extensive experiments demonstrate that the proposed methods are able to effectively invert the interactive pose from exemplar images and apply it to the customized generation with user-specified interaction.

## CCS CONCEPTS

• **Computing methodologies** → **Computer vision**; **Natural language processing**; **Machine learning approaches**.

## KEYWORDS

Textural Inversion, Customized Text-to-image Generation, Diffusion Model.

## 1 INTRODUCTION

Diffusion-based text-to-image generation models like Stable Diffusion (SD) [31] and DALL-E 2 [30] have shown great success in high-quality and diverse visual content generation. In order to meet user's customized requirements, there have been a series of excellent works [10, 21, 32, 36] proposed that investigate subject-driven generation based on pre-trained text-to-image diffusion models. They could facilitate many valuable applications, such as personalized portrait photos, virtual try-on, and art design.

Most subject-driven image generation methods either optimize a token embedding for a specific subject [1, 10, 36] or fine-tune the

Permission to make digital or hard copies of all or part of this work for personal or classroom use is granted without fee provided that copies are not made or distributed for profit or commercial advantage and that copies bear this notice and the full citation on the first page. Copyrights for components of this work owned by others than the author(s) must be honored. Abstracting with credit is permitted. To copy otherwise, or republish, to post on servers or to redistribute to lists, requires prior specific permission and/or a fee. Request permissions from permissions@acm.org.

*ACM MM, 2024, Melbourne, Australia*

© 2024 Copyright held by the owner/author(s). Publication rights licensed to ACM.
ACM ISBN 978-x-xxxx-xxxx-x/YY/MM
https://doi.org/10.1145/nnnnnnn.nnnnnnn

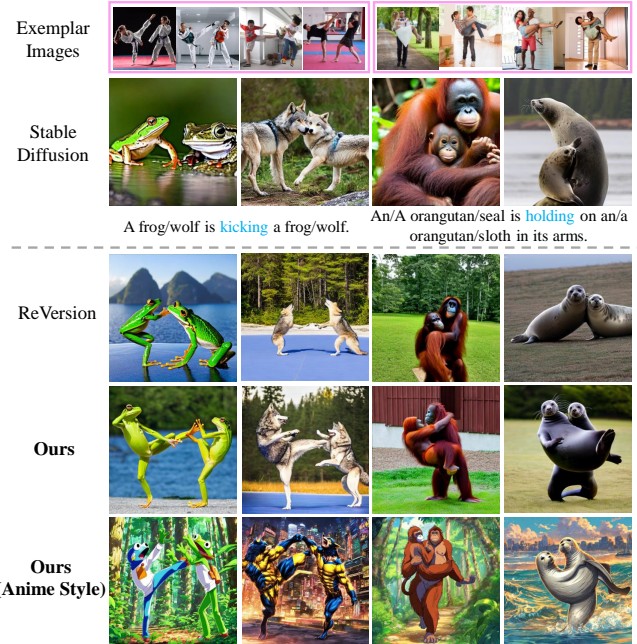

**Figure 1: Comparison between our method and existing T2I methods on generation with customized interactive pose. Compared to SD [31] and ReVersion [18], our method is able to produce reasonable customized generation results in both natural and animation styles.**

whole model or an image encoder on a set of images [8, 24, 32, 37]. With these optimized token embeddings or model weights, the specified subject can then be used as a new 'word' in text-to-image generation with either the pre-trained or the fine-tuned diffusion models. In addition, existing subject-driven image generation works are not limited to user-specified objects, but also work with other customized concepts, such as style [2, 36, 48], layout [1, 47] and action [15, 17]. Several works further extend subject customization from a single subject to multiple user-specified subjects [16, 18, 21, 23, 38]. However, most of these works focus on the modeling of each individual subject, while ignoring the relation or interaction between subjects. Interaction between subjects is very common in daily life and customized interaction between subjects has great potential in movie and animation making and artistic creation. However, it is very challenging to model the interaction between subjects with existing subject-driven generation methods.

*How can we customize the interaction between subjects in the text-to-image generation?* To address this task, Huang *et al.* makes an attempt in ReVersion [18] by designing a relation-steering contrastive learning strategy to the relation between subjects into a token embedding. However, that method, mainly relying on the preposition knowledge for the learned token embedding, specializes in learning spatial relations but could fail to give reasonable results in modeling interaction between subjects, as shown in Fig. 1. In addition, there is a category generalization issue when transferring

a specific kind of interaction for a certain category of object to other categories. To model the complex interaction between subjects, we could decompose it into several components and take a divide-and-conquer strategy. In addition, additional prior information could be leveraged to encourage the model to only learn to invert the interaction without binding to the specific appearance of these subjects.

Towards text-to-image generation with customized interactive pose, we propose a two-stage framework for customized interaction inversion from exemplar images. It adopts a divide-and-conquer strategy, which first computes a dedicated pseudo-word token to model a single pose for each individual subject in the interaction and then leverages them to further learn the concept for the interactive pose. Then, the interaction can be modeled as the combination of pseudo-word tokens for single poses and interactive pose. Specifically, to model the single pose in the first stage, we first extract the skeleton map [40] for each subject in the interaction and generate several images using ControlNet [46] with each one containing a single subject. Each image is described in the form of "$O[P]$", where $[P]$ is a pseudo-word for the pose of the subject $O$. To learn the concept of pose for each subject, we incorporate language prior in the inversion of that pose by collecting a list of verbs related to interactions and computing the mean and variance of their word embeddings. Then, the embedding for the pseudo-word $[P]$ is optimized to represent the single pose with the reconstruction objective. With the help of language prior, the learned embeddings of pseudo-words $[P]$ get close to the embedding space of verb. In the second stage, the embeddings of inverted single poses $[P^*]$ are employed to promote the learning of the interactive pose. The original exemplar images are described in the form of "$O_1[P_1^*][R]O_2[P_2^*]$". The pseudo-word $[R]$ for the interactive pose between $O_1$ and $O_2$ is optimized with specialized initialization from single poses. In addition to the reconstruction objective, a cross-attention loss is designed to highlight the interaction region. Once the learned token embeddings $[P_1^*]$, $[R^*]$ and $[P_2^*]$ are obtained, they can be inserted into any description as normal words for customized text-to-image generation.

The main contributions of this paper are summarized as follows.

- A divide-and-conquer strategy is proposed to achieve text-to-image generation with customized interaction.
- Language prior and a cross-attention loss are incorporated into the optimization process to promote the modeling of interaction.
- Extensive experiments demonstrate the effectiveness of the proposed method in customized generation with user-specified interaction.

## 2 RELATED WORK

**Text-to-Image Generation.** Generating images from natural language has provided great convenience to users. Some previous works based on GAN methods [27, 39, 45, 49] leverage text conditions to steer image generation, which synthetic images with high fidelity on domain-specific datasets. The auto-regressive generative model [26, 44] is another direction in the generative model, but these methods typically leads to inefficient image generation. Recently, a plethora of excellent work has emerged based on the diffusion model. Text-to-image (T2I) diffusion models [30, 31, 33] inject text into a unet-based diffusion model via pre-trained text encoders and cross-attention modules. Training on large-scale text-image pairs, diffusion models can yield high image-text alignment and image fidelity. Nevertheless, generating under complex text conditions remains challenging.

**Conditional Generation.** Some works [7, 28] observe diffusion model, such as Stable Diffusion, being constrained by the language understanding capacity of the text encoder, which result in the diffusion model struggle with complex prompts. To address this problem, Lian *et al.* [22] proposes LMD framework that enhances prompt understanding in text-to-image diffusion models through a novel two-stage generation process. Phung *et al.* [28] proposes attention-refocusing losses to regularize both attention layers during the sampling to improve the controllability given the layout and text prompt. Some works [4, 5, 12, 15, 25] address this problem with image editing. They encode the layout information in the cross-attention maps to control the layout of the generated image. In this paper, we focus on customizing specific interactive poses between subjects in the reference images, which otherwise would require plenty of words to describe that interaction in text-to-image generation. Different from the aforementioned methods, which either rely on the extra condition of the exemplar image or constrain the cross-attention maps, our method with texture inversion is more diverse and accessible.

**Customized Inversion.** In text-to-image tasks, some concepts may exhibit description ambiguities or be challenging to express in natural language. Formally, some user-specified concepts cannot be adequately captured using common tokens. Therefore, the task of capturing target concepts from user-provided images has raised widespread interest. DreamBooth [32] binds rare new words with specific subjects through fine-tuning the whole T2I generator. Kumari1 *et al.* [21] train a model to quickly acquire a new concept via closed-form constrained optimization. Recently, some works [3, 10, 21, 31] attempt to learn a token embedding for the subject's concept inversion. Then, generate a new customized image by using a new prompt in the form of "$[*]$ dog in the snow", where '$[*]$' is the optimized token embedding. Hamazaspyan *et al.* [11] presents DEPM that generates images based on the styles extracted from exemplar images. Zhang *et al.* [48] introduces InST to learning the high-level textual descriptions of a single painting image and then guiding the text-to-image generative model in creating images of specific artistic appearance. P+ [36] is proposed for further control of learned token in text-to-image generation by extending the textual-conditioning space with per-layer tokens. ADI [17] makes progress in learning specific action token from exemplar images with a single subject. Although it shows promising performance in capturing the action of a single subject, the training of each image with DreamBooth [32] would be time consuming. More recently, Huang *et al.* [18] proposes relation-steering contrastive loss to model the relation between subjects. It works well across various kinds of relations, such as spatial relationship, material and affiliation. However, the design of the method, which specializes in learning spatial relations, makes it challenging to model multiple concepts for an interaction in which the pose of each subject and the interaction between these poses are both important.

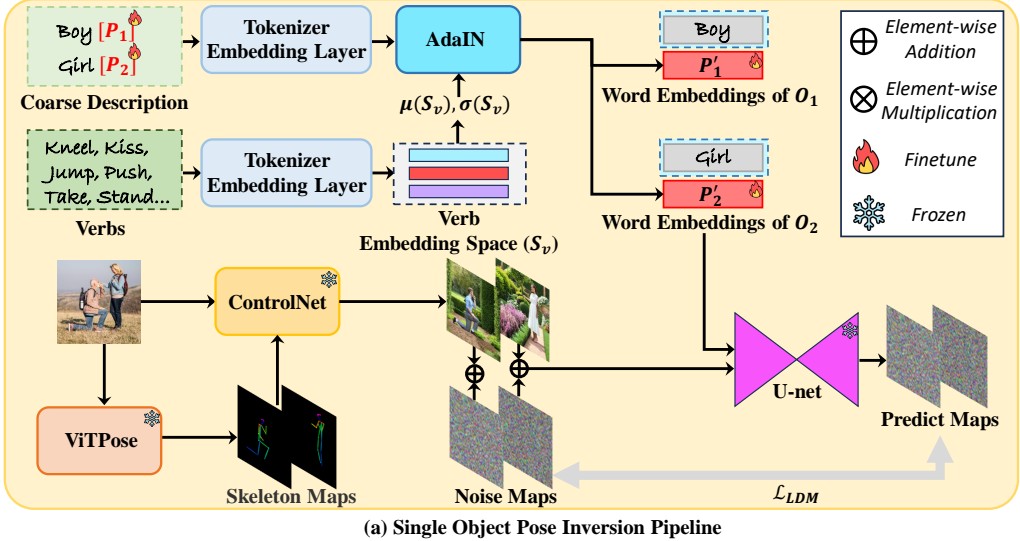

(a) Single Object Pose Inversion Pipeline

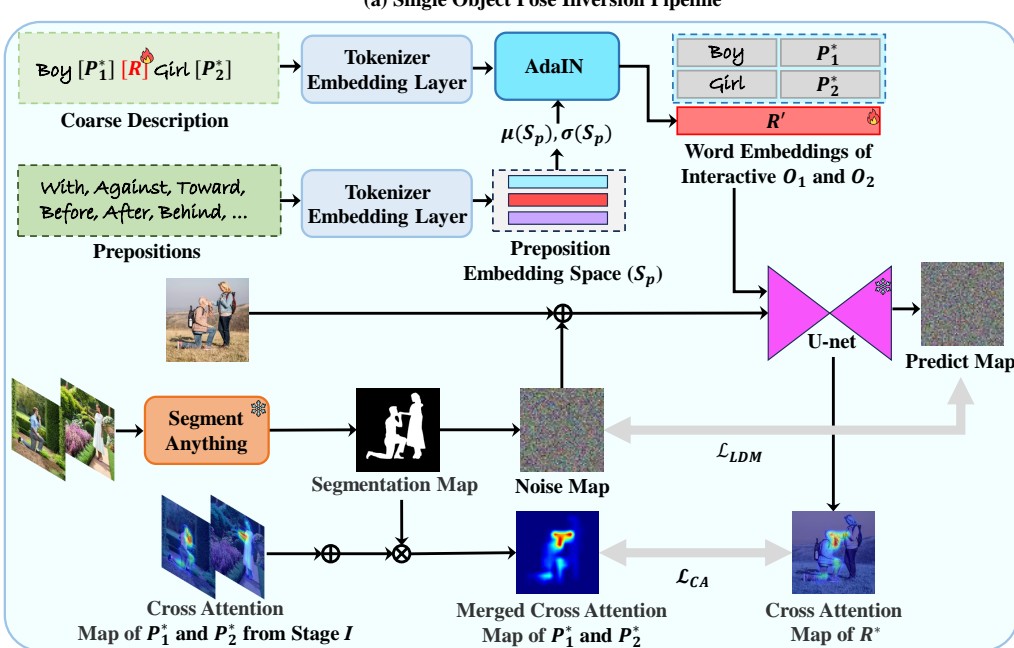

(b) Interactive Pose Inversion Pipeline

Figure 2: Framework for interaction-driven text-to-image generation. Given exemplar images and their coarse descriptions, the proposed method takes a divide-and-conquer strategy to learn interaction concepts in two stages, that is, single subject pose inversion (a) and interactive pose inversion (b).

## 3 PRELIMINARY

### 3.1 Latent Diffusion Model

Diffusion Model (DM) [13] is a type of generative model that learns the distribution of data by gradually denoising the noise sampled from the Gaussian distribution. Instead of operating on pixel space, the Latent Diffusion Model (LDM) [31] enhances performance by conducting denoising on latent space. It includes an autoencoder pre-trained on a large dataset and a conditional diffusion model. The Encoder $\mathcal{E}$ maps an image $x$ to a spatial latent code $z_0 = \mathcal{E}(x)$ where conditional diffusion model is applied, while the decoder

$D$ works the other way around. For text-to-image generation, a pre-trained text encoder encodes the text description '$y$' as $c_\theta(y)$, which is then injected into cross-attention layers of a U-Net model in the denoising process for conditional generation. The $\mathcal{L}_{\text{LDM}}$ training objective is formulated as:

$$\mathcal{L}_{\text{LDM}} := \mathbb{E}_{z \sim \mathcal{E}(x), y, \epsilon \sim \mathcal{N}(0,1), t} \left[ \|\epsilon - \epsilon_\theta (z_t, \mathbf{c}_\theta(y), t)\|_2^2 \right], \quad (1)$$

where $z_t$ is the latent variable noised to time $t$, starting from $z_0$, and $\epsilon_\theta$ is the denoising network. During inference, a sampled Gaussian noise $z_T$ is iteratively denoised to compute a new image latent variable $z_0$, which is transformed to image space by decoder

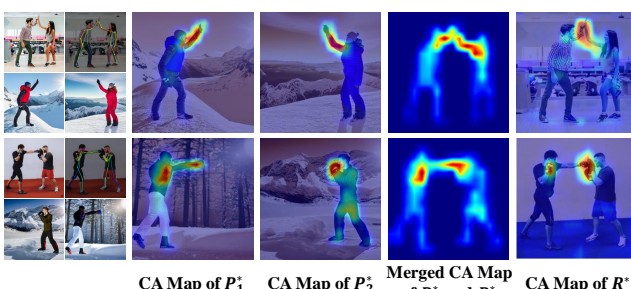

CA Map of $P_1^*$    CA Map of $P_2^*$    Merged CA Map of $P_1^*$ and $P_2^*$    CA Map of $R^*$

**Figure 3: Visualization of cross-attention maps in the interactive pose inversion.**

$x = D(z_0)$. Recently, Stable Diffusion (SD) [31] employs a cross-attention mechanism to inject textual conditions into the diffusion generation process, aligning with the provided textual input.

## 3.2 Textual Inversion

One category of methods for the subject-driven image generation task is to assign a special token to the specific subject and optimize the personalized token embedding. The optimization objective is defined as below:

$$v_* = \arg\min_v \mathbb{E}_{z,y,\epsilon,t} \left[ \| \epsilon - \epsilon_\theta \left( z_t, c_\theta(y, v), t \right) \|_2^2 \right], \quad (2)$$

which keeping both $\mathbf{c}_\theta(y)$ and $\epsilon_\theta$ fixed. The reconstruction objective motivates the learned embedding to capture visual concepts.

## 4 METHOD

### 4.1 Overall

In this task, there is a small set of exemplar images $\mathcal{I} = \{I_1, I_2, ..., I_n\}$ with a common user-specified interaction between subjects. In addition, each exemplar image is accompanied by a coarse description in the form of "$O_1^i [R] O_2^i$", where $O_1^i$ and $O_2^i$ denote the first and second subject tokens in the $i - th$ exemplar image respectively, $[R]$ represents a pseudo-word corresponding to the user-specified interaction. The aim of this task is to learn the concept of such specific interaction and transfer it into the process of text-to-image generation.

To address the task of text-to-image generation with customized interactive pose, we adopt a divide-and-conquer strategy and design a two-stage inversion framework, as shown in Fig. 2. The interaction between subjects $[R]$ is expanded into three pseudo-words $[P_1]$, $[R]$ and $[P_2]$ (here we mainly consider the case of interaction with two subjects). The introduction of $[P_1]$ and $[P_2]$ enhances the concept extraction for the single pose individual subjects. In this case, each image would be described in the form of "$O_1 [P_1] [R] O_2 [P_2]$". The first stage of inversion focuses on learning the token embedding $P_1^*$ and $P_2^*$ for single poses pseudo-word $[P_1]$ and $[P_2]$, as shown in Fig. 2 (a). In the second stage, the token embeddings of single poses are employed to promote learning the inversion of interactive pose $[R]$, as shown in Fig. 2 (b). We will detail these two stages in the following subsections.

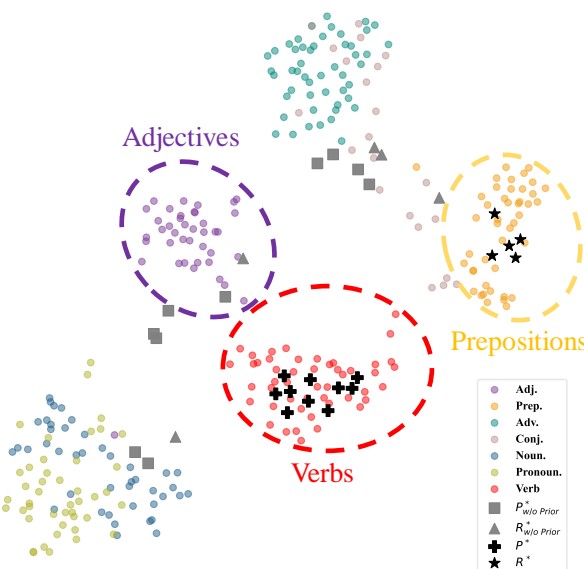

**Figure 4: Visualization of token embeddings learned with and without using the language prior by t-SNE [35].**

### 4.2 Single Subject Pose Inversion

To invert the single pose for each individual in the interaction, we aim to obtain images with each containing only one subject while maintaining the original pose of each subject as the exemplar image. Specifically, we adopt ViTPose+[41] to estimate the skeleton map of each subject in the interaction. Then, the ControlNet[46] refers to the estimated skeleton map to generate several images, with each one containing a single subject. The generated image with a new appearance of the subject and background is further used to learn the token embeddings $P_1^*$ and $P_2^*$ for single poses pseudo-word $[P_1]$ and $[P_2]$.

Inspired by the observation in [18] that the words of the same Part-of-Speech are closely clustered together, we incorporate the prior of the verb embedding space into the single pose inversion. Specifically, a list of verbs corresponding to common interactive action is either borrowed from predicates in [42, 43] or collected by ourselves. For example, this includes action verbs such as *stand, kneel and jump*. Text embeddings of these words together construct an embedding subspace for action tokens denoted as $S_v$. Then, we calculate the mean and variance of the subspace $S_v$ that is $\mu(S_v)$ and $\sigma(S_v)$, repetitively. The calculated mean and variance can be incorporated as a prior of verb embedding space into the original token embedding $P_1$ and $P_2$ to encourage token embedding learning. Specifically, we employ AdaIN [9] to incorporate the obtained verb prior into the token embedding $P_1$ and $P_1$:

$$P_i' = \sigma(S_v) \left( \frac{P_i - \mu(P_i)}{\sigma(P_i)} \right) + \mu(S_v), \ for \ i = 1, 2 \quad (3)$$

where $P_i'$ represents the embedding of token $[P_i]$ with the prior of verb embedding space. $\mu(P_i)$ and $\sigma(P_i)$ are scalars. $\mu(S_v) \in \mathbb{R}^d$ and $\sigma(S_v) \in \mathbb{R}^d$. The optimization goal in single subject pose inversion

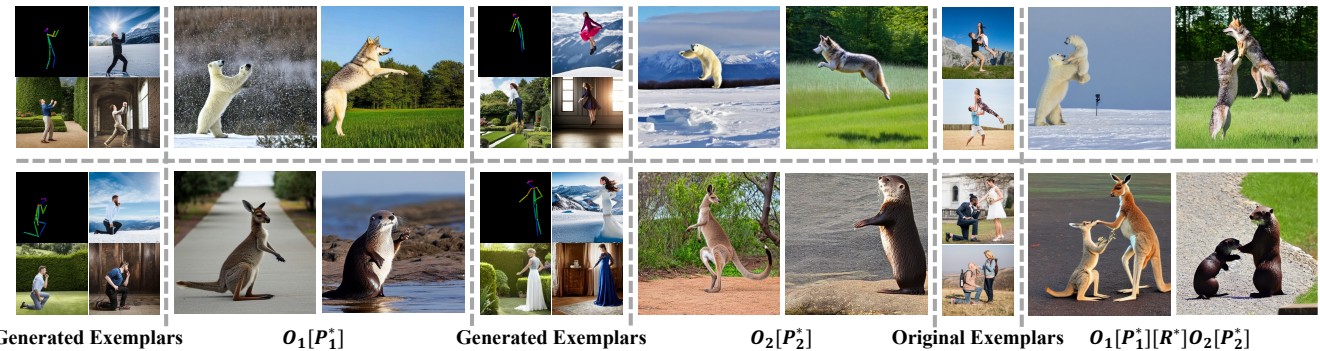

**Generated Exemplars**    $O_1[P_1^*]$    **Generated Exemplars**    $O_2[P_2^*]$    **Original Exemplars**    $O_1[P_1^*][R^*]O_2[P_2^*]$

**Figure 5: Results of text-to-image generation with various inverted concepts in both stages.**

stage can be defined as:

$$P_i^* = \arg\min_{P_i'} \mathbb{E}_{z,y,\epsilon,t}\left[\left\|\epsilon - \epsilon_\theta\left(z_t, c_\theta(y, P_i'), t\right)\right\|_2^2\right], \; for \; i = 1, 2 \quad (4)$$

where $P_i^*$ is the optimized embedding of trained token $[P_i^*]$. More details of the verbs used in this paper can be found in the appendix.

### 4.3 Interactive Pose Inversion

In order to model the interaction between subjects, we further introduce another pseudo-word $[R]$ between subjects $O_1$ and $O_2$ with learned pose token $[P_1^*]$ and $[P_2^*]$ in the second stage. We inverse token $[R]$ based on original complete exemplar images and fix trained tokens $[P_1^*]$ and $[P_2^*]$ of the first stage in the Sec. 4.2. Similar to the first stage, a list of prepositions is collected, which includes the commonly used words placed after verbs such as *toward, with, to, over and through*. We define the text embedding space of prepositions as $S_p$. Then, the preposition prior is calculated by:

$$R' = \sigma(S_p)\left(\frac{R - \mu(R)}{\sigma(R)}\right) + \mu(S_p), \quad (5)$$

where $R'$ represents the embedding of token $[R]$ with the prior of preposition embedding space. $\mu(R)$ and $\sigma(R)$ are scalars. $\mu(S_p) \in \mathbb{R}^d$ and $\sigma(S_p) \in \mathbb{R}^d$.

Different from the single subject pose inversion stage, we also aim to enhance our attention to the areas where interactions occur in the image. As mentioned in several works [12], cross-attention (CA) maps have a great influence on the spatial layout of objects in generated images. Therefore, we introduce guidance information from the cross-attention map between image patches and the pose concepts $[P_1^*]$ and $[P_2^*]$ of modeling interactive poses. Since the single subject pose in the first stage represents a part of the interaction, and focused regions are related to motion and pose in the cross-attention maps of a learned token $[P_1^*]$ and $[P_2^*]$. As shown in Eq. 6 and Fig. 3, we simply merge the two cross-attention maps by summing them together and filtering irrelevant background region:

$$G_{z_t} = \text{Norm}\left(\left(CA(P_1^*, z_t) + CA(P_2^*, z_t)\right) \times Mask, \quad (6)$$

where $CA$ calculate normalized cross-attention maps between the inverted single pose token $[P_1^*]$ and $[P_2^*]$ and noise latent $z_t$ at time step $t$. To avoid the influence of background, subjects' segmentation results [20] of the generated image corresponding to $z_t$ of the Stage I is computed compose the *Mask* and are applied to the merged

attention map. The $G_{z_t}$ is employed to regularize the cross-attention map for the token $[R']$ and the cross-attention loss is computed as below:

$$\mathcal{L}_{CA} = \mathbb{E}_{z_t}\left[\left\|CA\left(R', z_t\right) - G_{z_t}\right\|_2^2\right], \quad (7)$$

where pixel-wise mean squared error is used as the objective. Overall, the total loss for optimizing the token embedding of the interactive pose $[R]$ consists of the reconstruction objective as Sec. 3.2 and the additional cross-attention loss $\mathcal{L}_{CA}$:

$$R^* = \arg\min_{R'}\left(\mathbb{E}_{z,y,\epsilon,t}\left[\left\|\epsilon - \epsilon_\theta\left(z_t, c_\theta(y, R'), t\right)\right\|_2^2\right] + \lambda \mathcal{L}_{CA}\right). \quad (8)$$

where $R^*$ is the optimized embedding of trained token $[R]$. As for the loss weight $\lambda$, we found that $\lambda$ with a large value, such as 0.1 and 1, would cause a degradation in the appearance of subjects in the generated image. Therefore, we empirically set loss weight $\lambda_1$ to 0.01 in order to balance guidance from pixel and cross-attention map. In generation, the learned token $[P_1^*]$, $[R^*]$, and $[P_2^*]$ can be inserted into any description for text-to-image generation with customized interaction.

**Analysis.** Inspired by [12], we visualize the cross attention map between the exemplar image and the token embedding of the corresponding pseudo-word $[P_1^*]$, $[P_2^*]$ and $[R^*]$ in Fig. 3. It shows that the learned token can indeed attend to the region corresponding to the key parts of the concept of interactive pose. A similar phenomenon is observed in [34], which shows the map attends to key parts of human body for the action verb embedding. For example, the hands and arms are highlighted for the corresponding interactive pose.

In order to better understand the motivation for incorporating language prior to the inversion of interaction, we provide a further analysis. Fig. 4 visualizes the embedding space. The learned embeddings $[P_1^*]$ and $[P_2^*]$ are closer to the verb embedding cluster, while the $[R^*]$ are closer to the preposition embedding cluster. Compared to that, the token embeddings without using the language prior to the optimization process are randomly distributed, which tend to be clustered in the inaccurate embedding space, such as 'Adjectives' and 'Conjunction'. We also present a qualitative comparison to further validate the effectiveness of incorporating language prior to the optimization process, as shown in Fig. 8.

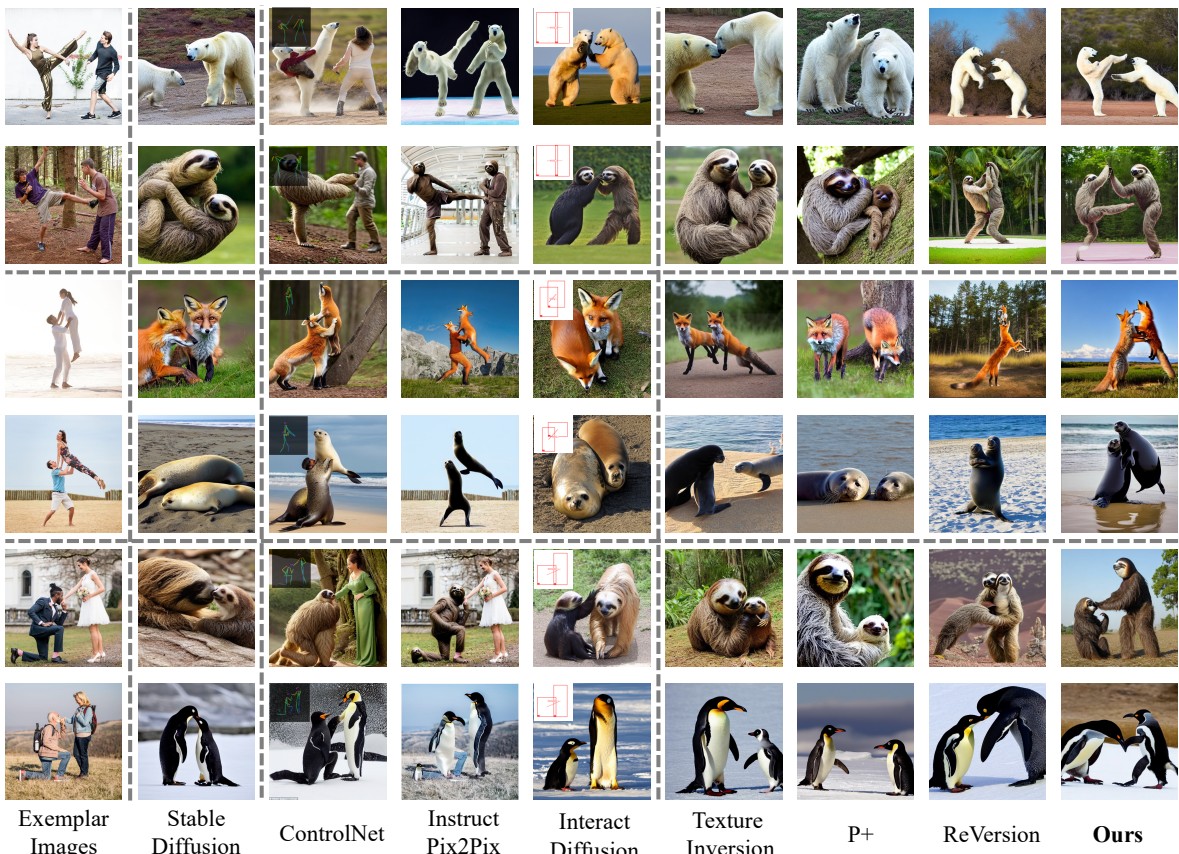

| Exemplar Images | Stable Diffusion | ControlNet | Instruct Pix2Pix | Interact Diffusion | Texture Inversion | P+ | ReVersion | **Ours** |

Figure 6: Qualitative comparison with existing methods. For SD, ControlNet, InteractDiffusion and InstructPix2Pix, we carefully design the input prompts according to the interaction in the exemplar images. For example, the prompts are "A ... is kicking a ...", "A ... is holding a ... above its head." and "A ... is kneeling down in front of a ..., holding its hand and kissing it.".

## 5 EXPERIMENT

### 5.1 Dataset

We collect 15 popular interactive actions from humans that happened in daily life, including *kneel down, kiss hands, hold in arms, ride on the back* and so on. For each of these interactions, we collect 10 diverse exemplar images by searching with a set of keywords on the internet. Each exemplar image contains two persons which is associated with a coarse description in a template form of "$[O_1]$ $[P_1]$ $[R]$ $[O_2]$ $[P_2]$". With such image and description pairs, our aim is to learn the interaction concept, that is, optimizing token embeddings for pseudo-word $[P_1]$, $[R]$ and $[P_2]$. Further introduction to the 15 interactive poses is presented in the appendix. *The collected data and code will be publicly available upon acceptance.*

**Evaluation Benchmark.** We construct a dataset for evaluation, which consists of 2,000 images generated by 200 descriptions, where each description generates 10 images. A total of 30 species categories are involved in these description.

### 5.2 Evaluation metrics

Different from subject-driven generation, it does not make much sense for us to generate images using the prompt "A photo of $[*]$" for interactive pose and compute the similarity between generated and exemplar images because it is difficult to describe a specific interactive pose with those few words and without mentioning the subjects. Therefore, we turn to large-scale pre-trained multimodal model CLIP [29] for help to evaluate open-domain interaction relation. Besides, we adopt the metrics *Pose-S* and *Pose-KP* for evaluating the generation accuracy of the subject pose, due to direct image similarity between generated images and exemplar images being interfered by subjects.

**CLIP-T.** We compute the similarity between the generated image and the description text (T) by computing a cross-modal matching score using CLIP. Here, we further extend the description text by replacing the pseudo-word with rich text, such as "A cat is kneeling down in front of a dog, holding its hand and kissing it".

**CLIP-S.** Similar to [10], we compute the subject similarity between the generated image and the description of "A photo of $[O_1]$ and $[O_2]$". Its purpose is to examine how the evaluated methods influence the generation quality of the subjects (S) in the description.

**Pose-S.** Similar to [18], we compute a classification accuracy for the generated pose of the subject (S) with a trained SVM classifier. To train an SVM classifier for the interactive pose, we first use the image encoder of the ViTPose+[41] to extract a 2048-dimensional feature vector of the exemplar image as the pose-related feature. Then, we adopt the extracted features to train an SVM classifier

Table 1: Quantitative comparison with existing inversion-based text-to-image generation models. The scores(%) of the CLIP-T, CLIP-S, Pose-S and Pose-KP are reported. Green text indicates the best and blue text indicates the second best performance.

| Method | CLIP-T↑ | CLIP-S↑ | Pose-S↑ | Pose-KP↑ |
|---|---|---|---|---|
| SD [31] | 23.51 | 21.87 | 7.05 | 6.36 |
| Texture Inversion [10] | 22.47 | 21.79 | 11.59 | 8.18 |
| P+ [36] | 20.55 | 22.19 | 12.42 | 9.23 |
| Reversion [18] | 24.68 | 22.81 | 18.47 | 15.53 |
| Ours | **25.74** | **22.93** | **40.81** | **38.64** |

with 15 classes. The trained SVM achieve 91% accuracy in the test set constructed by several example images different in the training process.

**Pose-KP.** This metric calculates the accuracy of the generated pose of subject based on key points (KP) of the estimated skeleton map. Specifically, we first train a two-layer GCN [19] with several exemplar images by using the key points of the skeleton map generated by the detector of ViTPose+[41]. The trained GCN achieves 85% multi-classification accuracy.

## 5.3 Comparison with Existing Methods

Since text-to-image generation with customized interactive pose is a novel task, there is a lack of research directly addressing it. In this section, we compare our methods with seven state-of-the-art methods,including, Stable Diffusion (SD) [31], ControlNet [46], Instruct-Pix2Pix [4], InteractDiffusion[14], Textural Inversion [10], P+[36] and ReVersion [18]. SD is a pure T2I method without using any additional conditions except for text. So, we take it as our baseline and use an extended description as its input for generation. Control-Net, InstructPix2Pix and InteractDiffusion are conditional-based generation methods. ControlNet refers to the provided skeleton map to generate the action. Note that, due to potential errors in OpenPose[6] caused by the occlusion of multiple individuals, we correct the results to avoid diminishing the performance. Since InstructPix2Pix is able to edit an image following user instruction, we adopt the exemplar image as its input. InteractDiffusion refers to the provided bounding boxes, which represent the location of the subject in the image as the condition. The last three are the inversion-based methods. Textual inversion introduces pseudo-word tokens and fine-tune embedding of them with reconstruction objective. P+ extends text condition, where different embeddings are injected into different layers of the U-net. ReVersion could revert relation to a token embedding for text-to-image generation. We train these inversion-based methods on our dataset and use the inverted concept for generation. Both qualitative and quantitative results are analyzed in this section.

**Qualitative Comparison.** The qualitative results are shown in Fig. 6. Although SD can generate correct subjects as the description mentioned, it fails to generate interactive poses as the exemplar images even though we provided a detailed description for it. The reason attributed to the user intent and machine understanding are not aligned. ControlNet just maintains a consistent pose with the exemplar image. It struggles to find a balance between the generated pose and the subject's appearance, resulting in incomplete or

Table 2: User-based quantitative results with competing methods. Interactive pose, subject and overall accuracy (%) are reported. Green text indicates the best and blue text indicates the second best performance.

| Method | Pose↑ | Subject↑ | Overall↑ |
|---|---|---|---|
| SD[31] | 16.4 | 78.3 | 11.7 |
| ControlNet [46] | 61.3 | 35.7 | 20.1 |
| InstructPix2Pix [4] | **90.8** | 29.3 | 23.4 |
| InteractDiffusion [14] | 34.5 | 62.8 | 21.3 |
| Inversion-based text-to-image generation models | | | |
| Texture Inversion [10] | 12.7 | 68.4 | 7.8 |
| P+ [36] | 15.2 | 74.5 | 9.3 |
| Reversion [18] | 29.1 | 84.9 | 24.2 |
| Ours | 58.6 | **85.6** | **48.9** |

distorted body structures. InstructPix2Pix also shows consistent pose with the target image, but it focuses too much on changing appearance of subjects rather than adapting the pose to the specified subject category. Therefore it would tend to generate subjects with blended appearances of different species. Based on additional layout conditions, InteractDiffusion controls the subject generated at the specified location but fails to generate interactive poses between subjects. This could be attributed to InteractDiffusion being trained on human-object interaction datasets, making it difficult to generalize to interactions between subjects. Additionally, the pure textual description may not be sufficient to specify complex interaction patterns. Inversion-based methods such as Textural Inversion and P+ have shown promising performance in subject-driven generation. However, these methods fail to decouple interaction concepts from subjects in exemplar images and generate inconsistent interactive poses. Reversion, which aims to model general relationships, struggles with complex interactive pose and tends to generate common simplified interactions such as hugging or shaking hands.

Our method with the proposed divide-and-conquer two-stage inversion strategy is more effective in extracting the concept of interactive pose between subjects and is able to flexibly adapt it to customized text-to-image generation tasks. Furthermore, we demonstrate that the proposed method also inherits the capability of Stable Diffusion in producing images in various styles. As shown in Fig. 7, our method can produce images in both natural style and animation style. More customized generation results for all involved interactive poses are shown in the appendix.

**Quantitative Comparison.** In the Tab. 1, we report quantitative results using proposed metrics mentioned in Sec. 5.2. Existing Inversion-based methods are not sufficient for extracting complex interactive poses and generate inconsistent interactive poses with worse pose accuracy(less than 20%). The proposed method achieves better balance with best performance on description consistency(CLIP-T), subject consistency(CLIP-S) and high pose accuracy(Pose-S, Pose-KP).

In addition, we also examine these methods through user study. Specifically, 20 human evaluators are asked to determine whether the generated interactive pose is consistent with those in the exemplar images as *Pose* accuracy and whether the generated subjects correspond with the description without obvious deformations or abnormalities as *Subject* accuracy. Furthermore, a comprehensive

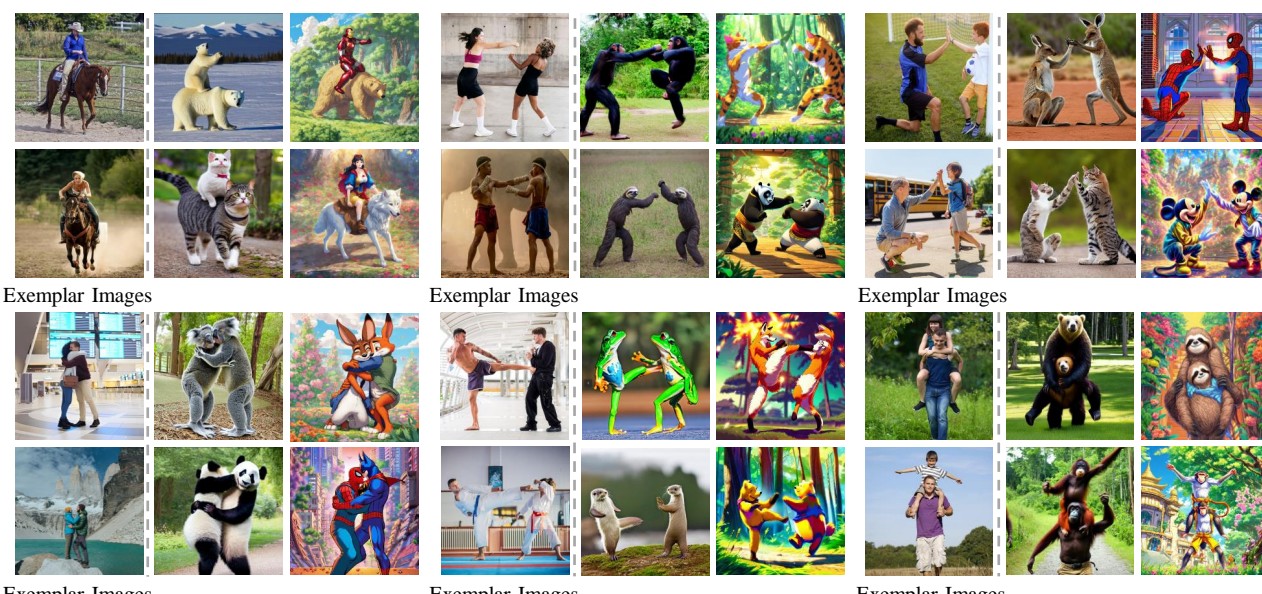

**Figure 7: Examples of our customized interaction generation.**

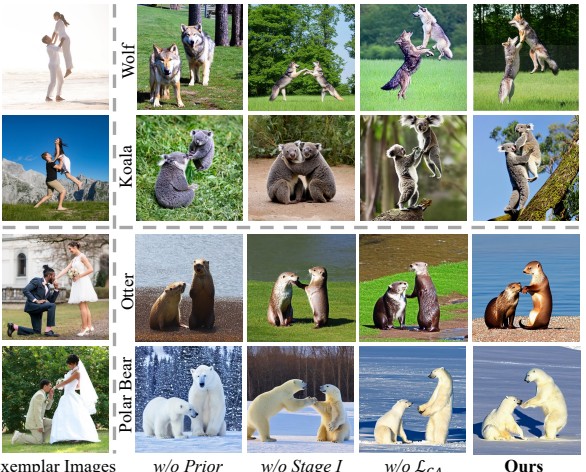

**Figure 8: Qualitative Comparisons with Ablation Variants.**

accuracy considering pose and subjects in the generated image. As shown in Tab. 2, pose-conditioned ControlNet and image-editing-based InstructPix2Pix struggle in generating subjects consistent with the description, thus achieve less then 40% *Subject* accuracy. InteractDiffusion and inversion-based baselines obtain less than 35% *Pose* accuracy, indicating difficulties in the generation of complex customized actions. The proposed method achieves 58.6% accuracy of interactive pose generation while maintaining high subject accuracy and improves overall accuracy by 24.7% compared to baseline methods similar to results reported in Tab. 1.

### 5.4 Ablation Study

To analyze the effectiveness of the proposed method, we also conduct an ablation study on the proposed components mentioned in

Sec. 4, and the results are shown in Fig. 8. For the baseline "*w/o Prior*", we optimize token $[P_1]$, $[P_2]$ and $[R]$ without incorporating language prior in Eq. 3 and Eq. 5. As in col2 of Fig. 8, there is a lack of relevant interaction between subjects in generated images, because of the failure to decouple the pose concept in the inversion stage. For the baseline "*w/o Stage I*", we skip the single subject pose inversion stage (Sec. 4.2) and directly modeling interactive pose as Sec. 4.3. Furthermore, we expand $[R]$ to three pseudo-words for fair comparison. Due to the lack of modeling for the single pose, the baseline can generate images with simplified and incorrect interactive poses, such as 'hug' and 'pounce', in col3 of Fig. 8. For the baseline "*w/o $\mathcal{L}_{CA}$*", we do not use cross-attention loss $\mathcal{L}_{CA}$ in the interactive pose inversion stage (Sec. 4.3). In col4 of Fig. 8, some details are missing in the key region where interaction or contact happens without cross-attention loss guidance. This issue could be mitigated by the proposed mask attention loss, which encourages the inversion to focus further on capturing the detailed interaction appearance.

## 6 CONCLUSION

In this work, we focus on customized generation with specific interactive poses between subjects from user-provided images. Interaction between subjects is very common in daily life and customized interaction between subjects has great potential in many application scenarios. A two-stage framework is proposed for customized interaction-driven generation. It adopts a divide-and-conquer strategy, which first extracts a dedicated text token for a single pose of each individual subject in the interaction and then leverages them to further infer the concept for the interactive pose. In addition, language prior and cross-attention loss are incorporated into the optimization process to promote the modeling of interaction. Extensive experiments demonstrate the effectiveness of the proposed method in customized generation with user-specified interaction.

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
