# OpenReview forum: "Customizing Text-to-Image Generation with Inverted Interaction"
_acmmm.org/ACMMM/2024/Conference — MM2024 Poster_

### Official Review · Reviewer_6N1t · 2024-05-20

**Rating:** 4
**Confidence:** 3

**Summary:**

The main contributions of this paper are summarized as follows: 1. A divide-and-conquer strategy is proposed to achieve text-to-image generation with customized interaction. 2. Language prior and a cross-attention loss are incorporated into the optimization process to promote the modeling of interaction. 3. Extensive experiments demonstrate the effectiveness of the proposed method in customized generation with user-specified interaction.

**Strengths:**

1. Customize the interaction between subjects in the text-to-image generation is useful in real-world applications.
2. Proposed divide-and-conquer strategy seems effective in this task, the experiments provided are sufficient.
3. Both the quantitative and qualitative performance of customized interaction in the paper show good results.

**Limitations:**

1. The customized strategy may limit the generation capabilities of VitPose and ControlNet.
2. More inverted tokens may limit the expressive ability of the model, especially in some complex texts.

Question:
1. Can you show more customized results with specific text descriptions? For example “O1[P1*][R*]O2[P2*] with Eiffel tower in the background”. In this paper, I can't confirm that the textual controllability of the model is well preserved.
2. Can you provide more detail about the Implementation of comparative baseline methods?
3. Intuitively speaking, simply finetuning ControlNet based on the extracted pose may also have a good result. Can you provide a comparative experiment?

**Suitability:**

3

---

### Official Review · Reviewer_YBvJ · 2024-05-24

**Rating:** 4
**Confidence:** 3

**Summary:**

This paper focus on transferring the interaction between subjects in exemplar images. It introduces a divide-and-conquer strategy to first learn the pose of each subject and then the interaction between subjects. It also propose integrating language priors and a unique cross-attention loss to enhance the modeling of the interaction. The quantitative and qualitative experiments validate the effectiveness of the proposed method.

**Strengths:**

- This paper addresses an interesting issue of how to learn the interaction between two subjects in an image, as opposed to focusing more on the individual characteristics of a subject. Interactions between subjects are relatively more abstract and difficult to learn.
- To better tackle this issue, the paper presents a divide-and-conquer approach, incorporating language priors and a specially designed cross-attention loss. The method is straightforward and easy to follow.
- Both the comparative experiments with other methods and the ablation studies seem to be comprehensive.

**Limitations:**

- What effect does the number of generated exemplars have on the results? How would the outcomes change if a similar process was used to create images with two objects present, bypassing the initial stage and using textual inversion right away? Is the main improvement in results due to the expansion of data volume or the methodology itself? BTW, if DreamBooth [1] employ these images for training, how will it perform? Although it is not an inversion-based text-to-image generation method, it remains a classical approach for the subject-driven area, which is similar to the problem at hand.
- What is the performance of this method when the image scale and resolution change?
- How would this method generalize to interactions between humans and objects, and what is its performance in multi-person scenarios? Additionally, what are the primary limitations of this method?

[1] Nataniel Ruiz, Yuanzhen Li, Varun Jampani, Yael Pritch, Michael Rubinstein, Kfir Aberman. DreamBooth: Fine Tuning Text-to-Image Diffusion Models for Subject-Driven Generation. CVPR 2023.

**Suitability:**

2

---

### Official Review · Reviewer_vjzA · 2024-05-27

**Rating:** 2
**Confidence:** 4

**Summary:**

Overall, while the manuscript presents an interesting research topic, the current version requires significant improvements in terms of novelty, experimental evaluation, clarity, and completeness. Therefore, I recommend rejecting the manuscript in its current form and encouraging the authors to address the below-mentioned issues before resubmitting it for consideration.

**Strengths:**

See below comments for details.

**Limitations:**

1. While the authors present an interesting approach for customizing text-to-image generation with interaction inversion, the novelty of the proposed method is not clearly demonstrated. The manuscript would benefit from a more thorough comparison with existing techniques to better highlight its unique features.
2. The experimental evaluation presented in the manuscript is insufficient to fully assess the performance of the proposed method. More comprehensive experiments, including ablation studies and comparisons with state-of-the-art approaches, should be conducted to provide stronger evidence of the effectiveness of the proposed technique.
3. The manuscript lacks clarity in some parts, particularly in the explanation of the technical details of the proposed framework. The authors should revise the manuscript to improve the readability and coherence of the text, especially in the methodology section.
4. The manuscript does not adequately discuss the limitations and potential drawbacks of the proposed method. A more balanced presentation of the strengths and weaknesses of the approach would enhance the overall quality of the manuscript.
5. The manuscript does not provide enough information about the datasets used in the experiments. The authors should include more detailed descriptions of the data sources and preprocessing steps to enable reproducibility of the results.
6. The manuscript does not sufficiently address the broader impact of the proposed method. The authors should discuss the potential implications of their work in terms of applications, societal effects, and ethical considerations.
7. Some figures and tables in the manuscript are poorly formatted and difficult to interpret. The authors should revise the visual elements to ensure that they are clear, concise, and informative.
8. The manuscript contains several grammatical errors and typos that detract from its overall quality. The authors should carefully proofread the manuscript to correct these issues before submitting it for publication.

**Suitability:**

3

---

### Official Review · Reviewer_rPJk · 2024-05-28

**Rating:** 4
**Confidence:** 3

**Summary:**

This paper introduces a novel method for customizing text-to-image generation with interaction inversion. The method adopts a two-stage interaction inversion framework, which first learns pseudo-words for individual subject poses using a divide-and-conquer strategy, and then uses these pseudo-words to further learn the concept of interactive poses. Language prior and cross-attention loss are incorporated into the optimization process to promote interaction modeling.

**Strengths:**

1. Relevant pipelines have been proposed for both single-object and multi-object scenarios.

2. The interaction between objects is achieved using an additional pseudo-word.

3. Experiments demonstrate that the proposed method can effectively invert the interactive poses from exemplar images and apply them to customized generation with user-specified interactions.

**Limitations:**

1. The limited learning of prepositions or verbs has led to numerous failure cases in the example images.

2. The proposed methods places significant demands on the quality of prompts.

**Suitability:**

3

---

### Meta-Review · Area_Chair_qjFY · 2024-07-03

**Recommendation:** Accept (Poster)
**Confidence:** 5

**Metareview:**

This paper introduces a novel method for customizing text-to-image generation with interaction inversion. This is a novel concept that customizes relationships while transferring targets, which has significant application and research value. The authors have provided a preliminary feasible solution for this setup and addressed most of the reviewers' concerns during the rebuttal process. After carefully reviewing the authors' paper and the reviewers' comments, I am confident that this work makes a valuable contribution to the community.